# Learning and Evaluating Visual Similarity Discovery under Incomplete Labeling

## Abstract

Visual Similarity Discovery (VSD) focuses on retrieving *positives*: images of distinct objects that exhibit perceptual similarity to a given query. This is a core need in applications like e-commerce and visual search. This work advances VSD research through several key contributions. First, we introduce a new VSD dataset in the furniture domain with over 63K labeled image pairs, providing a valuable resource for VSD learning and evaluation. Second, we propose two evaluation metrics that enable more reliable and consistent VSD performance assessment under incomplete labeling. Third, we show that supervised finetuning of multiple pretrained models on VSD labels significantly improves VSD performance. Finally, we present Soft Positive Augmentation, a method that leverages existing VSD labels to infer soft positive relations among unlabeled pairs via weighted graph transitivity. Augmenting the VSD labels with these inferred soft positives during finetuning yields additional performance gains. Our code and dataset will be made publicly available.

## 1 Introduction

Visual Similarity Discovery (VSD) addresses the challenge of retrieving images of distinct items that exhibit perceptual similarity to a given query image (Park et al., 2019; Douze et al., 2021; Barkan et al., 2023). In the VSD task, the system is provided with a query image and required to retrieve visually similar images, referred to as *positives*, from a closed catalog of candidate images. Importantly, the interest is on retrieving images of *different* and *distinct* items (typically products) from the one shown in the query, that still share a high degree of perceptual similarity as judged by humans. This task is central to applications such as visual search and recommender systems, where visually similar alternatives are suggested based on human-perceived similarity.

In this work, we consider a simple and straightforward VSD pipeline. We assume a model that is capable of computing similarity between image pairs (e.g., by producing embeddings for each image in a latent vector space, followed by applying a similarity function). Once similarities are computed, the image catalog is ranked w.r.t. the query image, and the top-K highest-ranked images are retrieved as the final results[1]. Then, further filtration is applied to ensure that all retrieved images correspond to different items. Specifically, if two or more retrieved images belong to the same catalog item, only the highest-ranked image is preserved as the representative of that item, while the others are removed from the retrieval list.

Recent studies highlight that VSD diverges from traditional tasks like object identification and recognition (Barkan et al., 2023; Sundaram et al., 2024). Unlike conventional classification and metric learning approaches, which rely on object identity or category for supervision (Razavian et al., 2016; Deng et al., 2019), VSD requires models to prioritize perceptual similarity, a requirement on which traditional methods have been shown to underperform (Barkan et al., 2023). These findings motivate the need for novel, human-annotated datasets specifically tailored for evaluating and training VSD models.

The Efficient Discovery of Similarities (EDS) method (Barkan et al., 2023) was the first to establish a benchmark for VSD in the fashion domain. It uses a set of models, called *generators*, to retrieve the top-K most similar images per query, which are then labeled by experts as similar (positive) or dissimilar (negative). Assuming the generators surface truly similar items with high probability, EDS significantly improves positive discovery rate compared to random sampling. However, EDS has a

---

[1]We note that other aspects of the retrieval system, such as the efficiency of the retrieval mechanism, often involving approximate nearest neighbor search algorithms, are out of scope for this work, as our focus is on learning and evaluating VSD models.

key limitation: since annotators are exposed only to the top-K results surfaced by the generators, the labeling process is inherently biased toward those generators. Consequently, evaluating a new model whose top-K results differ from the generators' becomes problematic, as its unique retrievals lack labels. This generator bias can lead top-K metrics to underestimate the performance of models that surface unlabeled yet relevant results. As a remedy, Barkan et al. (2023) proposed using ROC-AUC (AUC) for assessing VSD performance. Unlike top-K metrics, AUC evaluates the probability that a positive item ranks above a negative one, regardless of absolute rank, and was shown to provide more consistent evaluations in the presence of missing labels.

This work introduces a new VSD dataset and advanced methods for training and evaluating VSD models, particularly under incomplete labeling: First, we present a novel VSD dataset in the furniture domain, comprising 63,298 expert-labeled image pairs annotated as either similar (positive) or dissimilar (negative). To optimize the annotation process, we followed the EDS paradigm, which efficiently surfaces high-probability positive pairs, thereby improving the positive discovery rate. However, as previously noted, EDS inherently biases the dataset toward the generator models used during retrieval.

Therefore, we introduce two new evaluation metrics tailored for evaluating VSD in scenarios with incomplete labels. The first, Discounted Credit Score (DCS), enables more nuanced evaluation of ranking results by controllably emphasizing the importance of top-ranked retrievals. Importantly, DCS scores query-retrieval pairs individually, overcoming the triplet-based limitations of AUC. DCS is shown to outperform standard metrics such as AUC and BPREF (Buckley & Voorhees, 2004) across various consistency tests. The second metric, Estimated Hit-Ratio at K (EHR@K), approximates the true Hit-Ratio at K (HR@K) (Barkan et al., 2023) in cases where labels are missing among top-K retrievals. Our findings show that EHR@K correlates well with HR@K, offering a reliable estimate of model performance under incomplete labeling.

Finally, we highlight the advantage of supervised finetuning on VSD labels. To this end, we employ several seminal supervised losses, utilizing the VSD labels produced by the EDS method. We demonstrate that supervised finetuning of various pretrained models consistently improves VSD performance across VSD datasets and metrics. Moreover, we present Soft Positive Augmentation (SPA) - a method that utilizes the existing ground truth (GT) VSD labels to infer soft positives among unlabeled pairs. Then, the soft positives are used to augment the GT labels during the supervised finetuning process, providing additional performance boost. Overall, these finding provides evidence for the quality of the generated VSD labels and suggests that pretrained models, when supervised using VSD labels, are capable of learning representations that align with human-perceived similarity. The effectiveness of the proposed methods and metrics is empirically validated through extensive evaluations on two VSD datasets, establishing a new state-of-the-art benchmark in VSD research.

To summarize, our main claims and contributions are as follows: (1) A new VSD dataset in the furniture domain, offering a valuable resource for training and evaluating VSD models. (2) Two evaluation metrics designed for assessing VSD performance under incomplete labeling. (3) Empirical evidence that supervised finetuning on VSD labels significantly improves performance, with additional gains achieved through the SPA method.

## 2 RELATED WORK

Evaluating visual similarity is a complex challenge in content-based image retrieval (Eakins & Graham, 1999). Initiatives like the Image Similarity Challenge (ISC21) have advanced this by introducing fine-grained granularity schemes for defining similarity (Douze et al., 2021). However, many visual similarity models primarily rely on instance identification to assess whether different images depict identical items, with common challenges arising from variations in angles, lighting, and model appearances, as evident in datasets such as DeepFashion (Liu et al., 2016), Street2Shop (Liu et al., 2012), and DARN (Hadi Kiapour et al., 2015; Wang et al., 2016; Huang et al., 2015).

Unlike simple identification, visual discovery entails recognizing nuanced item resemblances that align with human perception, necessitating expert input. For example, Shankar et al. (2017) curated a proprietary, expert-annotated dataset tailored for such evaluations. This need for expert insights is also reflected in methodologies that utilize popular image search queries for annotations, which, however, may not always be appropriate for offline datasets (Wang et al., 2014).

In response to these challenges, recent advancements like the EDS method have emerged, providing the first VSD benchmark in the fashion domain (Barkan et al., 2023). Another line of work aims to capture human-like perceptions of visual similarity through neural networks trained on synthetically generated image triplets with human-annotated similarity ratings (Fu et al., 2023; Sundaram et al., 2024).

Our work takes VSD research a step forward, and further draws parallels to classic Information Retrieval (IR) studies such as the Cranfield experiments, which established the framework for large-scale evaluation of IR systems (Cleverdon, 1967). We address the contemporary issue of large, incompletely labeled datasets by developing new evaluation metrics that accommodate incomplete relevance assessments (Buckley & Voorhees, 2004; Moffat et al., 2007), thus enhancing the robustness and fairness of visual discovery evaluations. Moreover, unlike previous studies that primarily evaluate pretrained models for VSD tasks (Barkan et al., 2023), we propose leveraging available VSD labels produced by human annotators for supervised finetuning (Hadsell et al., 2006; Weinberger et al., 2005; Musgrave et al., 2020; Khosla et al., 2021). This approach is shown to consistently improve VSD performance, outperforming the original pretrained models across all VSD metrics and datasets.

## 3 A NOVEL VSD DATASET

We introduce VSD-Furniture, a novel VSD dataset comprising 63,298 labeled image pairs in the furniture domain. This dataset was curated by labeling image pairs as either similar or dissimilar within the furniture category of the publicly available Google Universal Image Embedding (GUIE) dataset[2]. The furniture category includes 10,458 images spanning diverse furniture types and styles. Representative examples can be explored on the GUIE dataset's web page. VSD-Furniture will be released under the CC0 (public domain) license, ensuring free and unrestricted use, consistent with the licensing of GUIE-Furniture.

The labeling process followed the EDS procedure (Barkan et al., 2023), which is designed to mine similar image pairs with high efficiency. Below, we provide a brief overview of EDS (for a more comprehensive description, the reader is referred to Barkan et al. (2023)).

Let $D$ represent the dataset of images, and let $Q \subset D$ be the set of query images. EDS employs a set of generator models $G$, where each model $m \in G$ provides a heuristic similarity score $S_m(a, b) \in \mathbb{R}$ for any image pair $(a, b) \in D$. For a given query $q \in Q$, each model $m$ ranks all images in $D$ by similarity to $q$, defining the *top-K retrievals* as the $K$ images most similar to $q$. For each query $q \in Q$, the top-K retrievals from all generator models are then aggregated into a set $H$ that contains all query-retrieval pairs.

Human annotators then assess each query-retrieval pair in $H$, labeling it as either similar (positive) or dissimilar (negative). The annotation task is distributed among $T$ expert annotators, with each pair in $H$ reviewed by at least two annotators to ensure consistency. Ambiguous cases are flagged for group discussion, allowing annotators to reach a consensus label or, if necessary, exclude the pair from $H$. This EDS procedure results in a high-quality, curated VSD dataset, $A = \{(a, b, y_{ab}) : (a, b) \in H\}$, where $y_{ab} \in \{0, 1\}$ represents the ground-truth (GT) label: 1 for positive, and 0 for negative.

For the VSD-Furniture dataset, we randomly sampled 3,494 queries from $D$ (GUIE-Furniture). We then employed the following four pretrained models as generators to retrieve the top-$K$ candidates (with $K = 5$) for each query: 1) Argus Vision (**AS**) - a ResNext101 model pretrained on Bing web data[3], 2) **DINO** (Caron et al., 2021) - a self-supervised model pretrained on ImageNet1K, 3) **BEiT** (Bao et al., 2021) - pretrained on ImageNet21K (Ridnik et al., 2021), and 4) **CLIP** (Radford et al., 2021) - an image encoder pretrained on web-scale data. The similarity function $S_m$ was set to the cosine similarity for all models. The resulting retrievals were aggregated into $H$ and subsequently sent for human annotation. The resulting VSD-Furniture dataset, after removing duplicates and excluding controversial pairs, consists of 63,298 labeled query-retrieval pairs, with 39,194 labeled as positive and 24,104 as negative. Additional details about the dataset, annotators, annotation process and guidelines, as well as labeled examples are provided in Appendix B.

## 4 THE DISCOUNTED CREDIT SCORE METRIC

A key limitation of the GT dataset $A$ (resulting from the EDS labeling process described in Sec. 3) is its bias towards the generators in $G$. This bias arises because expert annotators only review the top-K retrievals produced by specific the models (generators) in $G$. Consequently, when evaluating a new model $m \notin G$, top-K retrievals not included in $A$ lack corresponding labels. This scenario of incomplete labels presents challenges in assessing new models using top-K metrics, as some or all labels for top-K retrievals may be missing.

To address this issue, consistency tests were proposed in Barkan et al. (2023) to evaluate the reliability of metrics in scenarios with incomplete labels. These tests determine whether the ranking of models

---

[2]https://www.kaggle.com/datasets/rhtsingh/130k-images-512x512-universal-image-embeddings
[3]https://pypi.org/project/argusvision/

based on VSD performance remains consistent when using a given metric, regardless of whether the labels are complete or incomplete. The results showed that top-K metrics often fail these consistency tests. As an alternative, Barkan et al. (2023) proposed the AUC metric, which is not a top-K metric and has demonstrated robust performance in these tests. AUC evaluates performance across the entire ranking spectrum, beyond just the top-K retrievals, by calculating the probability that a positive retrieval is ranked higher than a negative one throughout the set $A$. Other related metrics such as BPREF (Buckley & Voorhees, 2004), follows a similar approach.

Despite its robustness, the AUC metric has certain limitations. In search or recommendation applications, the quality of top-K retrievals is paramount. The AUC penalty for a negative in the top ranks is linear in the number of positives ranked below it. Moreover, AUC is indifferent to the absolute ranks of positive and negative pairs, focusing solely on their relative ranking, whether the positive is ranked higher than the negative. As a result, AUC can only assess pairs of retrievals and lacks the ability to assign a score based on absolute rank. Practically, the significance of a negative retrieval appearing in the top-5 far outweighs a similar occurrence in the top-100 to top-200 range.

To this end, we introduce the Discounted Credit Score (DCS) metric. DCS operates on the percentile rank of a retrieval and behaves **differently** based on whether the retrieval is labeled as positive or negative. DCS scores individual retrievals by considering both their absolute ranking and label. It is designed to credit (penalize) positives (negatives) ranked at the top percentiles and penalize (credit) positives (negatives) ranked at the bottom percentiles. The percentile regime considered as the bottom is controlled by an adjustable continuous parameter. DCS is defined as follows:

$$C(p, y, \alpha) = \phi(p, \alpha)^y (1 - \phi(p, \alpha))^{1-y}, \quad (1)$$

with

$$\phi(p, \alpha) = \frac{\exp(\alpha p) - 1}{\exp(\alpha) - 1}, \quad (2)$$

where $p \in [0, 1]$ and $y \in \{0, 1\}$ represent the percentile rank and the label (1 for positive and 0 for negative) of the retrieval, respectively. The parameter $\alpha \in (0, \infty]$ controls the shape of the credit curve and the range of $p$ considered as the bottom.

Figure 1 illustrates DCS curves ($C$ values as a function of $p$) for different $\alpha$ values, for positive ($y = 1$) and negative ($y = 0$) cases. $C$ ranges from 0 to 1, reaching 0 and 1 for $p = 0$ and $p = 1$, respectively. As $\alpha$ approaches 0, $C$ linearly credits higher (lower) $p$ for positive (negative) retrievals. With increasing $\alpha$, the credit 1) grows (vanishes) exponentially for positives (negatives) in the highest $p$ regime, and 2) collapses to a constant 0 (1) for positives (negatives) outside that regime. Essentially, higher $\alpha$ values emphasize the top percentiles, dividing the credit curve into distinct top and bottom regimes. Conversely, when $\alpha$ is near 0, DCS shows linear growth (decay) across the entire $p$ spectrum. Therefore, the $\alpha$ parameter enables flexible adjustment based on the importance assigned to top-ranked retrievals. Note that different $\alpha$ values can be used for positive and negative labels depending on specific goals and business requirements. In this work, we use $\alpha = 10$ for both positive and negative retrievals. Our experiments show that DCS outperforms both AUC and BPREF metrics in various consistency tests, confirming its effectiveness as a reliable measure in scenarios with incomplete labels.

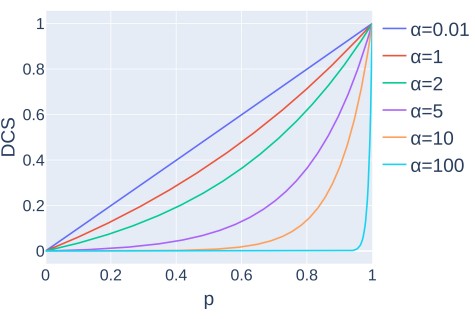

(a) Positive label ($y = 1$) scores

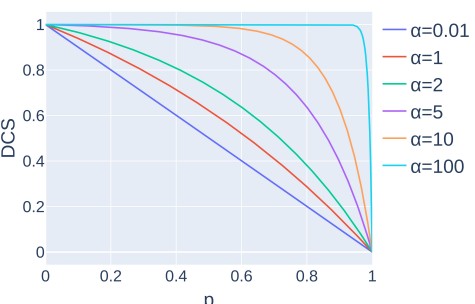

(b) Negative label ($y = 0$) scores

Figure 1: DCS function curves for various choices of $\alpha$.

# 5 THE ESTIMATED HIT-RATIO AT K METRIC

In information retrieval applications, the quality of rankings is often judged based on the relevance of the top-K items, as these are the most visible to users. Metrics that focus on top-K items are crucial because they closely reflect real-world user experiences when comparing different ranking models.

However, calculating popular top-K metrics such as HR@K, MRR@K, and NDCG@K, becomes challenging when labels for the top-K items are missing (Buckley & Voorhees, 2004). While metrics like BPREF and AUC can be used in such cases, they serve as suboptimal proxies for top-K metrics and do not focus on the top-K retrievals.

This work centers on the HR@K metric[4], defined as follows: Given a query and a list of top-K retrievals, HR@K is the number of positive retrievals divided by K. It is important to note that in other studies, HR@K may be defined differently, often aligning with Precision@K. For the purposes of this discussion, we use the definition provided here.

A naive approach to report HR@K in the presence of missing labels among the top-K retrievals is to count the number of positively labeled items within the top-K and divide that count by K. However, this approach can unfairly penalize models when labels for their top-K retrievals are unknown. For example, a model that retrieves only positive items in the top-K would receive an HR@K of 0 if all labels for these retrievals are missing.

To address this issue, we propose an adaptation of the HR@K metric for scenarios with incomplete labels, termed Estimated HR@K (EHR@K). EHR@K aims to provide a more accurate estimate of the true HR@K in cases where some labels are missing. Instead of using a constant denominator K, EHR@K divides the number of positive retrievals in the top-K by the number of labeled retrievals in the top-K.

We also introduce the concept of coverage@K, which represents the fraction of top-K retrievals that have labels. When coverage@K equals 1, EHR@K matches HR@K exactly, as all top-K retrievals are labeled. However, when coverage@K is 0 (indicating that all labels for the top-K retrievals are missing), we propose setting EHR@K to the average EHR@K across all examples with coverage@K > 0 for that specific model. If a model consistently retrieves unlabeled items in the top-K, we recommend abandoning top-K metrics in favor of alternatives like AUC, BPREF, or the proposed DCS metric for model evaluation.

In Sections 7.2.1 and 7.2.2, we evaluate the proposed EHR@K metric through a series of consistency tests, demonstrating its effectiveness in estimating top-K performance in scenarios with incomplete labels.

## 6 SUPERVISED VSD FINETUNING

Barkan et al. (2023) focused on VSD evaluation using pretrained models. In this work, we contend that a pretrained model $m$ can benefit from subsequent finetuning on the labeled VSD dataset $A$ produced by the EDS method. To this end, we investigate whether the use of VSD labels improves performance across several seminal supervised representation learning losses: Triplet loss (**TRPL**) (Weinberger et al., 2005), Contrastive loss (**Con**) (Hadsell et al., 2006), and Supervised Contrastive loss (**SupCon**) (Khosla et al., 2021).

While numerous alternative losses have been proposed over the last decade, their improvements over TRPL and Con losses, when evaluated under rigorous experimental protocols, have been found to be marginal at best (see Fig. 3 in Musgrave et al. (2020)). Therefore, we consider these three foundational losses sufficient to provide comprehensive empirical evidence for the effectiveness of using VSD labels, should they yield performance improvements when used to finetune a variety of pretrained models.

We further examine the effect of finetuning pretrained models on the dataset images using the seminal self-supervised learning methods **DINO** Caron et al. (2021) and **SimCLR** (Chen et al., 2020), without incorporating VSD labels directly into the loss function. Instead, the VSD labels are used only by the VSD metric for hyperparameter tuning and for monitoring model performance during training. As we show in Sec. 7, these methods produce mixed results that are inferior to supervised finetuning and, in some cases, even worse than using the pretrained model without finetuning. Yet, we emphasize that our claim centers on the benefit of leveraging VSD labels in supervised learning to improve VSD performance. We do **not** aim to make a general comparison between supervised and self-supervised learning for VSD tasks, as that is beyond the scope of this work.

Throughout all finetuning experiments, including both self-supervised and supervised methods, we maintained a consistent experimental framework, with deviations noted only when necessary. All methods were evaluated on identical dataset splits (train, validation, and test), and the code

---

[4]While the issue of computing top-K metrics with incomplete labels applies broadly to metrics such as HR@K, MRR@K, NDCG@K, etc. this work specifically examines HR@K as a case study, reserving the investigation of other metrics for future research.

implementation will be released on GitHub. Implementation details (preprocessing, optimization process, hyperparameter tuning, model architecture, similarity computation, etc.) are provided in Appendix E.

## 6.1 SOFT POSITIVE AUGMENTATION

To further enhance the supervised finetuning process, we propose applying Soft Positive Augmentation (SPA). SPA augments the GT VSD label set $A$, with newly inferred soft positive relations between unlabeled pairs based on the existing labels in $A$. These inferred soft positives are integrated with $A$ and used for supervised finetuning of model $m$. Specifically, SPA assigns a soft positive score in the range $[0, 1]$ to each unlabeled image pair. These scores are computed via a function that estimates the *positiveness* of each pair using weighted graph transitivity over the labeled pairs in $A$. SPA is compatible with any supervised representation learning loss that supports soft labels, e.g., by weighting the loss function accordingly. Our empirical evaluation reveals that SPA improves performance across all evaluated supervised losses and VSD metrics. In what follows, we describe the SPA method in detail (using the notation introduced in Sec. 3).

SPA begins by constructing an undirected graph where images in $D$ serve as nodes. In this graph, edges between nodes $(a, b)$ are weighted as 1 for positive pairs (i.e., $(a, b) \in H$ and $y_{ab} = 1$), and $\infty$ for negative (i.e., $(a, b) \in H$ and $y_{ab} = 0$) or unlabeled pairs (i.e., $(a, b) \notin H$). Then, a shortest-paths algorithm is applied to this graph, with a maximal distance $L$ (i.e., distances larger than $L$ are considered infinite, disconnecting those nodes). The algorithm outputs a distance function $d_{ab} \in \{0, 1, \dots, L, \infty\}$, where $d_{ab} = 0$ if and only if $a = b$. Then, the *positiveness* of an image pair $(a, b)$ is defined as:

$$\mathcal{P}(a, b) = \begin{cases} y_{ab} & \text{if } (a, b) \in H, \\ \exp(-\beta d_{ab}) & \text{if } (a, b) \notin H, \end{cases} \quad (3)$$

where $\beta$ is a hyperparameter controlling the rate of exponential decay. The positiveness function in Eq. 3 adheres to the original label $y_{ab}$ for the GT labeled pairs $(a, b) \in H$, while for unlabeled pairs $(a, b) \notin H$, $\mathcal{P}(a, b) \in [0, 1]$ is determined by the exponential decay based on $\beta$ and $d_{ab}$. Thus, shorter distances $d_{ab}$ yield higher positiveness scores. Notably, when $a = b$, we have $d_{ab} = 0$, resulting in $\mathcal{P}(a, b) = 1$. In our experiments, setting $L = 7$ and $\beta = 0.7$ produced the best results across all metrics and datasets, on average. in Appendix C, we present evaluations that ablate on the design choices in SPA (e.g., the softness nature of $\mathcal{P}$, and its hyperparameters).

Equipped with the positiveness function $\mathcal{P}$, one can assign a positiveness score to any pair $(a, b)$. When finetuning with SPA, all in-batch pairs for which $\mathcal{P}(a, b) > 0$ are treated as positive pairs, weighted by their positivesness scores. For example, to apply SPA to the SupCon loss (following the notation in Eq. 2 of SupCon (Khosla et al., 2021)), we augment the original positive set of sample $i$, denoted $P(i)$ (note the distinction from $\mathcal{P}$), to include all $p$ for which $\mathcal{P}(i, p) > 0$. Then, each log term in Eq. 2 of (Khosla et al., 2021) is weighted by $\mathcal{P}(i, p)$, and the inner sum is divided by $\sum_{p \in P(i)} \mathcal{P}(i, p)$ instead of $|P(i)|$. The application of SPA to the TRPL and Con losses proceeds in the same manner.

## 7 EXPERIMENTAL SETUP AND RESULTS

All experiments were executed on an NVIDIA DGX machine equipped with 4×A100 GPUs, using the PyTorch framework.

### 7.1 DATASETS, MODELS, AND METRICS

Results are reported for two VSD datasets: Fashion (Barkan et al., 2023) and Furniture (our newly proposed dataset). For consistency, we used the same set of generators used to form the Fashion dataset (Barkan et al., 2023), hence the generators are the same for both datasets as described in Sec. 3. In addition, we evaluated the performance of four non-generator models: DINOv2 (**DINO2**) (Oquab et al., 2023), OpenCLIP (**OC**) (Ilharco et al., 2021), and SWAG (**SWAG**) (Singh et al., 2022). Together, the evaluation encompasses seven different models.

Evaluation metrics vary across experiments, with **DCS**, **EHR**, **HR**, BPREF (**BPF**) (Buckley & Voorhees, 2004), and ROC-AUC (**AUC**) Macro and Micro (Barkan et al., 2023) used to assess metric consistency and correlation, while finetuning performance was measured using DCS, AUC, and EHR. Following the findings of Buckley & Voorhees (2004); Barkan et al. (2023), we exclude the Mean Average Precision (MAP) metric from our evaluation, as it has been shown to be less effective than AUC and BPF in scenarios with incomplete labels. While the evaluation in Barkan et al. (2023)

included traditional top-K metrics (HR@K and MRR@K), this was primarily to demonstrate their ineffectiveness in scenarios with incomplete labels, as discussed in Sec. 5. Therefore, in our work, HR is used solely as a benchmark to assess the performance of other VSD metrics under the ideal full coverage (fully labeled) scenario (See Experiment 2). Additional details regarding the datasets and metrics are provided in Appendices B and F, respectively.

Both Furniture and Fashion datasets underwent a split at the image level into 75% for training and and 25% for testing. Additionally, for training monitoring and hyperparameter optimization purposes, we created a separate validation set from the training data. This split adhered to a training:validation ratio of 80% : 20%.

Results for all methods (supervised, self-supervised, and pretrained) are reported on the test set. Statistical significance was tested using a paired t-test, confirming that the differences between the best performing supervised method and the best among the pretrained and self-supervised methods are statistically significant with $p < 0.05$.

## 7.2 EXPERIMENTS

We conduct experiments to validate the effectiveness of: (1) the proposed DCS and EHR metrics in evaluating and comparing VSD model performance under incomplete labeling, and (2) supervised finetuning using VSD labels, with or without SPA.

### 7.2.1 EXPERIMENT 1

This experiment aims to evaluate how consistent (i.e., insensitive to generator bias) are the proposed VSD metrics. To this end, we followed the leave-one-out experimental settings used in previous consistency evaluations from Barkan et al. (2023), designed to test the sensitivity of VSD metrics to generator bias. In this approach, the annotations for the top retrievals surfaced by each generator model are excluded from the dataset in turn. This process allows us to measure the impact of omitting each generator's annotated data (in turn) on the evaluation metrics, providing insights into potential biases within these metrics. The **Score** column in Tab. 1 reflects the average and standard deviation of the tested VSD metric scores for each generator model across all four possible leave-one-out subsets.

In this experiment, we define the evaluation results for each metric as the list of the metric scores obtained by all models when evaluated on the dataset (e.g., for the DCS metric it is simply the mean DCS score obtained by all models when evaluated on the dataset). Metric consistency is quantitatively assessed by measuring the correlation between the evaluation results (list of metric scores obtained by each model) produced in two different setups: one using the full dataset (with the full set of annotations) and another using a reduced dataset in which all annotated retrievals from the generator under examination are excluded. We consider three correlation measures: Spearman correlation (**SC**), Kendall's Tau (**KT**), and Pearson correlation (**r**). A *high* correlation indicates that the metric is *less* sensitive to generator bias, as the scores or ranking of the evaluated models remain consistent regardless of the exclusion.

The three correlation scores for this experiment are reported under the **Bias** section for both the Fashion and Furniture datasets in Tab. 1. On the Furniture dataset, we observe that the DCS metric demonstrates the highest consistency, followed by AUC and EHR, suggesting it is less prone to bias when a generator's data is excluded. On the Fashion dataset, DCS, EHR, and AUC metrics perform similarly on average. Across both datasets, the least consistent performer is BPF.

### 7.2.2 EXPERIMENT 2

Beyond the consistency tests suggested in Barkan et al. (2023), we further examine the correlation between each VSD metric's evaluation results and those of the HR@5 metric in the ideal 'full coverage' scenario, i.e., where **all** retrievals for a query are annotated. This complements the previous consistency experiment by assessing whether a metric is not only robust to generator bias but also produces evaluation results that align with HR@K, which is considered the ideal metric in fully labeled scenarios.

To this end, we repeat the leave-one-out experiment but consider a subset of queries for which **all** top-5 retrievals are annotated. We then measure consistency using the same correlation measures as in the original consistency test, but instead of evaluating the self-consistency of VSD metrics under the omission of generator data, we compute their correlation with the HR@5 scores obtained by the models. This allows us to assess whether the VSD metrics align with the ideal top-K metric (HR@5) in a **fully** labeled scenario.

| Metric | Model | Fashion | | | | | | Furniture | | | | | | |
|---|---|---|---|---|---|---|---|---|---|---|---|---|---|---|
| | | Score | Bias | | | FC | | | Score | Bias | | | FC | | |
| | | | SC | KT | r | SC | KT | r | | SC | KT | r | SC | KT | r |
| DCS | DINO | $78.97 \pm 1.22$ | 1.0 | 1.0 | 0.99 | 0.8 | 0.67 | 0.75 | $63.24 \pm 3.17$ | 0.97 | 0.89 | 0.94 | 1.0 | 1.0 | 0.99 |
| | CLIP | $75.62 \pm 0.96$ | 1.0 | 1.0 | 1.0 | 0.94 | 0.85 | 0.57 | $62.82 \pm 2.62$ | 0.98 | 0.94 | 0.98 | 0.8 | 0.67 | 0.88 |
| | BEiT | $82.58 \pm 0.98$ | 1.0 | 1.0 | 1.0 | 0.8 | 0.67 | 0.7 | $65.17 \pm 3.36$ | 0.85 | 0.72 | 0.8 | 0.6 | 0.33 | 0.6 |
| | AS | $73.23 \pm 1.2$ | 1.0 | 1.0 | 0.99 | 0.94 | 0.84 | 0.84 | $65.83 \pm 3.16$ | 0.9 | 0.78 | 0.83 | 0.88 | 0.84 | 0.84 |
| EHR@5 | DINO | $93.28 \pm 0.44$ | 1.0 | 1.0 | 1.0 | 0.8 | 0.67 | 0.6 | $70.37 \pm 4.15$ | 0.93 | 0.83 | 0.91 | 0.94 | 0.84 | 0.98 |
| | CLIP | $90.76 \pm 2.57$ | 0.93 | 0.83 | 0.83 | 0.8 | 0.67 | 1.0 | $64.76 \pm 8.34$ | 0.83 | 0.67 | 0.88 | 0.8 | 0.67 | 0.88 |
| | BEiT | $95.18 \pm 0.68$ | 0.98 | 0.94 | 0.98 | 0.94 | 0.85 | 0.62 | $77.09 \pm 3.84$ | 0.85 | 0.67 | 0.95 | 1.0 | 1.0 | 0.96 |
| | AS | $89.47 \pm 2.62$ | 0.95 | 0.89 | 0.81 | 0.8 | 0.67 | 0.84 | $74.29 \pm 3.28$ | 0.94 | 0.82 | 0.94 | 0.8 | 0.67 | 0.96 |
| $AUC_{mic}$ | DINO | $69.44 \pm 0.72$ | 0.98 | 0.94 | 0.88 | 0.8 | 0.67 | 0.83 | $65.55 \pm 1.72$ | 0.9 | 0.78 | 0.99 | 0.8 | 0.67 | 0.8 |
| | CLIP | $67.62 \pm 1.92$ | 0.88 | 0.78 | 0.98 | 0.94 | 0.85 | 0.96 | $65.2 \pm 2.49$ | 0.92 | 0.78 | 0.97 | 0.8 | 0.67 | 0.68 |
| | BEiT | $74.83 \pm 0.67$ | 0.85 | 0.78 | 0.93 | 0.8 | 0.67 | 0.8 | $74.14 \pm 2.98$ | 0.82 | 0.67 | 0.98 | 0.6 | 0.33 | 0.67 |
| | AS | $62.56 \pm 2.06$ | 0.93 | 0.83 | 0.97 | 0.4 | 0.33 | 0.5 | $72.03 \pm 2.12$ | 0.85 | 0.82 | 0.78 | 0.84 | 0.78 | 0.78 |
| $AUC_{mac}$ | DINO | $74.0 \pm 1.03$ | 1.0 | 1.0 | 0.99 | 0.8 | 0.67 | 0.77 | $67.59 \pm 2.28$ | 0.93 | 0.83 | 0.98 | 0.94 | 0.84 | 0.82 |
| | CLIP | $71.65 \pm 2.51$ | 0.87 | 0.72 | 0.95 | 0.95 | 0.89 | 0.97 | $61.26 \pm 3.98$ | 0.85 | 0.67 | 0.87 | 0.88 | 0.84 | 0.84 |
| | BEiT | $78.86 \pm 0.79$ | 0.89 | 0.92 | 1.0 | 0.8 | 0.67 | 0.76 | $73.54 \pm 3.54$ | 0.97 | 0.89 | 0.97 | 0.8 | 0.67 | 0.95 |
| | AS | $67.85 \pm 2.8$ | 0.83 | 0.78 | 0.88 | 0.4 | 0.33 | 0.25 | $71.45 \pm 2.22$ | 0.9 | 0.83 | 0.98 | 0.94 | 0.87 | 0.93 |
| BPF | DINO | $30.38 \pm 5.85$ | 0.78 | 0.67 | 0.85 | 0.4 | 0.33 | 0.25 | $35.99 \pm 10.6$ | 0.68 | 0.56 | 0.69 | 0.6 | 0.33 | 0.29 |
| | CLIP | $21.2 \pm 4.6$ | 0.88 | 0.78 | 0.91 | 0.0 | 0.0 | 0.1 | $29.46 \pm 7.75$ | 0.8 | 0.67 | 0.89 | 0.4 | 0.33 | 0.99 |
| | BEiT | $30.34 \pm 5.53$ | 0.83 | 0.78 | 0.87 | 0.67 | 0.67 | 0.3 | $41.9 \pm 11.15$ | 0.77 | 0.61 | 0.66 | 0.2 | 0.0 | 0.26 |
| | AS | $22.27 \pm 5.13$ | 0.72 | 0.61 | 0.89 | 0.4 | 0.33 | 0.78 | $38.26 \pm 11.32$ | 0.5 | 0.5 | 0.64 | 0.4 | 0.33 | 0.04 |

Table 1: Consistency evaluation produced by a leave-one-out experiment on different metrics. See Secs. 7.2.1 and 7.2.2 for details.

The correlation scores are reported under the full coverage (**FC**) section in Tab. 1. The results demonstrate the effectiveness of DCS and EHR, which exhibit strong correlations competitive with the AUC metric. These findings complement the consistency tests, showing that our newly proposed DCS and EHR metrics not only remain stable under the exclusion of generator data but also correlate well with HR@5 in fully labeled scenarios, reinforcing their reliability. A comprehensive analysis considering the correlation of the VSD metrics across various different coverage levels is presented in Appendix D.

### 7.2.3 EXPERIMENT 3

This experiment aims to evaluate whether supervised finetuning of pretrained models using VSD labels improves VSD performance, with or without SPA. Table 2 presents VSD metric results across all backbone models and finetuning methods for the Furniture and Fashion datasets. Due to hardware constraints, DINO finetuning results are reported only for the DINO2 and BEiT backbones. The results for the pretrained (non-finetuned) models are denoted as **Pre**. When SPA is applied to a supervised method, it is indicated by adding a SPA subscript to the method name.

We observe the following trends: all supervised finetuning methods, with or without SPA, contribute to improved performance across all VSD metrics and datasets, outperforming both the pretrained versions and the self-supervised finetuning methods. In addition, applying SPA improves performance of all supervised methods.

Among the supervised approaches, SupCon$_{SPA}$ emerges as the best-performing method on average. The second-best performer alternates between Con$_{SPA}$ and TRPL$_{SPA}$, with no clear winner between them. In particular, for the DCS metric, SupCon$_{SPA}$ and TRPL$_{SPA}$ are the top-performing methods on the Fashion and Furniture datasets, respectively. Nevertheless, in most cases, the performance differences between SupCon, TRPL, and Con are modest, and their overall effectiveness is arguably comparable. The similarity in performance between Con and TRPL aligns with prior findings (Musgrave et al., 2020). SupCon's comparable performance is also expected, as it is itself a contrastive loss that generalizes TRPL by supporting multiple positives and negatives per anchor.

By contrast, the self-supervised methods underperform relative to their supervised counterparts and, in many cases, perform on par with or even worse than the pretrained model. Among these, there is no clear winner: while SimCLR and DINO perform similarly on the EHR metric, SimCLR achieves better results on the AUC and DCS metrics. Notably, DINO exhibits degradation on these specific metrics compared to the original pretrained model. This suggests a fundamental difference between the VSD task and the conventional self-supervised learning paradigms investigated here. While self-supervised methods primarily focus on aligning representations of different augmentations of the same instance, the VSD task involves learning relations between distinct items based on human perceptual similarity, requiring a more nuanced understanding of inter-instance relationships.

**(a) Furniture finetuning results.**

| Backbone | Method | EHR | | AUC | | DCS |
|---|---|---|---|---|---|---|
| | | @5 | @20 | mac | mic | |
| SWAG | SupCon$_{SPA}$ | **88.19** | **83.67** | **74.8** | **77.54** | 69.78 |
| | Con$_{SPA}$ | 85.98 | 83.07 | 73.5 | 75.76 | 69.44 |
| | TRPL$_{SPA}$ | 86.71 | 82.27 | 74.78 | 76.4 | **69.8** |
| | Con | 85.51 | 82.19 | 73.58 | 75.21 | 69.0 |
| | TRPL | 86.59 | 81.34 | 74.05 | 75.39 | 69.01 |
| | SupCon | 83.92 | 80.79 | 74.4 | 76.27 | 68.48 |
| | SimCLR | 79.45 | 77.2 | 69.53 | 69.67 | 66.47 |
| | Pre | 80.76 | 78.16 | 69.68 | 69.01 | 65.54 |
| OC | SupCon$_{SPA}$ | **88.23** | **83.76** | **75.35** | **78.27** | 69.97 |
| | Con$_{SPA}$ | 86.55 | 83.53 | 74.35 | 76.65 | 69.6 |
| | TRPL$_{SPA}$ | 86.68 | 83.17 | 74.79 | 77.23 | **69.98** |
| | Con | 85.69 | 82.36 | 74.42 | 75.79 | 69.15 |
| | TRPL | 85.93 | 81.86 | 74.71 | 76.29 | 68.66 |
| | SupCon | 84.84 | 81.38 | 74.74 | 77.03 | 68.57 |
| | SimCLR | 83.36 | 80.43 | 71.41 | 72.51 | 67.61 |
| | Pre | 83.09 | 80.32 | 71.53 | 71.35 | 67.57 |
| DINO2 | SupCon$_{SPA}$ | **88.13** | 83.74 | **75.49** | **78.41** | 70.22 |
| | Con$_{SPA}$ | 87.78 | **83.89** | 73.89 | 76.27 | 69.54 |
| | TRPL$_{SPA}$ | 86.7 | 83.07 | 75.26 | 77.06 | **70.38** |
| | Con | 84.93 | 82.3 | 73.85 | 75.6 | 68.97 |
| | TRPL | 85.77 | 82.16 | 74.98 | 76.83 | 69.07 |
| | SupCon | 85.28 | 81.04 | 74.66 | 76.88 | 68.67 |
| | SimCLR | 84.36 | 80.25 | 72.88 | 72.5 | 68.28 |
| | DINO | 84.88 | 80.66 | 69.93 | 68.27 | 63.38 |
| | Pre | 85.14 | 80.38 | 72.27 | 72.38 | 67.65 |
| BEiT | SupCon$_{SPA}$ | **88.82** | **83.87** | 75.39 | **77.68** | 69.59 |
| | Con$_{SPA}$ | 86.81 | 83.51 | 73.67 | 75.82 | 69.22 |
| | TRPL$_{SPA}$ | 86.44 | 82.39 | **75.4** | 76.77 | **69.83** |
| | Con | 85.16 | 81.98 | 73.25 | 74.71 | 68.62 |
| | TRPL | 86.09 | 82.03 | 75.34 | 76.58 | 68.48 |
| | SupCon | 86.22 | 80.8 | 75.11 | 76.61 | 68.28 |
| | SimCLR | 84.41 | 80.46 | 74.07 | 73.19 | 68.2 |
| | DINO | 79.32 | 75.89 | 62.97 | 58.88 | 59.94 |
| | Pre | 84.53 | 80.18 | 72.15 | 70.7 | 67.73 |

**(b) Fashion finetuning results.**

| Backbone | Method | EHR | | AUC | | DCS |
|---|---|---|---|---|---|---|
| | | @5 | @20 | mac | mic | |
| SWAG | SupCon$_{SPA}$ | **99.37** | 98.8 | **92.79** | **86.63** | **87.89** |
| | Con$_{SPA}$ | 98.94 | 98.7 | 92.26 | 85.37 | 87.37 |
| | TRPL$_{SPA}$ | 99.27 | **98.84** | 91.91 | 82.98 | 87.08 |
| | Con | 98.5 | 98.67 | 91.99 | 83.33 | 84.99 |
| | TRPL | 98.65 | 98.4 | 91.76 | 81.94 | 86.31 |
| | SupCon | 99.19 | 98.52 | 92.22 | 83.65 | 85.19 |
| | SimCLR | 98.6 | 97.83 | 87.92 | 75.01 | 79.17 |
| | Pre | 97.71 | 97.29 | 87.14 | 75.68 | 75.62 |
| OC | SupCon$_{SPA}$ | **99.41** | **98.86** | **92.6** | **86.48** | **87.81** |
| | Con$_{SPA}$ | 99.3 | 98.81 | 92.18 | 85.09 | 85.11 |
| | TRPL$_{SPA}$ | 99.24 | 98.58 | 91.76 | 85.23 | 87.35 |
| | Con | 98.68 | 98.71 | 92.05 | 83.71 | 84.91 |
| | TRPL | 99.15 | 98.52 | 91.35 | 81.74 | 86.63 |
| | SupCon | 98.93 | 98.48 | 92.06 | 83.53 | 87.06 |
| | SimCLR | 99.11 | 98.64 | 89.75 | 78.55 | 82.55 |
| | Pre | 99.1 | 98.49 | 90.25 | 80.72 | 82.85 |
| DINO2 | SupCon$_{SPA}$ | **99.26** | 98.77 | **92.87** | **86.57** | **87.74** |
| | Con$_{SPA}$ | 99.0 | 98.67 | 92.27 | 84.8 | 85.19 |
| | TRPL$_{SPA}$ | 99.25 | **98.94** | 91.68 | 85.19 | 87.38 |
| | Con | 99.17 | 98.7 | 92.03 | 83.56 | 84.87 |
| | TRPL | 98.73 | 98.42 | 91.56 | 82.29 | 86.51 |
| | SupCon | 99.03 | 98.34 | 92.13 | 83.5 | 87.19 |
| | SimCLR | 97.2 | 96.22 | 84.99 | 69.32 | 76.74 |
| | DINO | 98.16 | 97.0 | 83.65 | 64.07 | 57.79 |
| | Pre | 97.3 | 96.87 | 86.14 | 70.1 | 69.42 |
| BEiT | SupCon$_{SPA}$ | 98.69 | 98.37 | **92.0** | **84.93** | **87.58** |
| | Con$_{SPA}$ | **98.97** | 98.37 | 91.52 | 83.06 | 87.09 |
| | TRPL$_{SPA}$ | 98.52 | 97.91 | 91.19 | 83.08 | 86.91 |
| | Con | 98.13 | 97.94 | 90.99 | 81.26 | 84.07 |
| | TRPL | 98.31 | 97.88 | 90.47 | 80.42 | 86.32 |
| | SupCon | 98.36 | 98.12 | 91.43 | 81.85 | 84.45 |
| | SimCLR | 96.61 | 95.89 | 86.97 | 74.46 | 82.3 |
| | DINO | 95.04 | 95.03 | 79.42 | 54.71 | 43.91 |
| | Pre | 96.03 | 95.43 | 87.85 | 75.67 | 83.85 |

Table 2: Finetuning results across all backbones, methods, and metrics for (a) Furniture and (b) Fashion datasets. See Sec. 7.2.3 for details.

In addition, we observe that VSD performance across all models is consistently higher on the Fashion dataset than on the Furniture dataset (See Tab. 2). This disparity may be attributed to the increased difficulty of the Furniture dataset, which sometimes features more complex and varied backgrounds with distractor elements. Such scenes make it more challenging for models to isolate and focus on the target objects, thereby complicating the perceptual similarity assessment.

We further observe a clear trend in the behavior of the evaluation metrics: the EHR metric yields the highest absolute scores across models, followed by the AUC metric, with DCS typically producing the lowest scores. This pattern reflects the varying levels of strictness embedded in each metric and the different aspects of performance they are designed to capture in missing labels scenarios.

Overall, the findings in Tab. 2 indicate the following: (1) The pairwise human-annotated VSD labels produced via the EDS paradigm contain meaningful signals, enabling effective learning and generalization of human-perceived visual similarity. (2) Supervised VSD finetuning of pretrained models improves performance, and combining it with SPA leads to further gains.

# 8 CONCLUSION

This work advanced VSD research on multiple fronts. First, we introduced a new VSD dataset in the furniture domain, comprising 63K labeled image pairs. We hope this dataset will serve as a valuable resource for accelerating the development and evaluation of VSD models. Second, we proposed two new metrics designed to support VSD evaluation under incomplete labeling. Our experiments show both metrics to be robust and effective, particularly in real-world scenarios where missing labels are common. Third, we demonstrated the benefits of supervised finetuning on VSD labels, showing performance improvements. Finally, we introduced the SPA method, which infers soft positive relations between unlabeled pairs and incorporates them into the finetuning process, yielding additional gains. By establishing this benchmark, we aim to drive continued progress in VSD research, fostering advancements in model development and evaluation. Due to space constraints, we discuss limitations and directions for future work in Appendix G.

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
