## A APPENDIX OVERVIEW

The appendix provides detailed information and supplementary results complementing the main paper. Specifically, Appendix B provides details on the two VSD datasets used in this work, including details on the annotators and annotation process of the VSD-Furniture as well as examples of labeled image pairs.

In Appendix C, we present extensive ablation studies justifying the design choices taken in the implementation of SPA in this work. Appendix D complements Experiment 2 (Sec.7.2.2) by examining the correlation between VSD-metric evaluation results across various coverage levels, and HR@5 results in the ideal full coverage scenario.

Appendix E provides additional implementation details of the evaluated representation learning methods, including preprocessing, optimization process, hyperparameter tuning, model architecture, and similarity computation. Appendix F describes the metrics used in our experiments. Finally, Appendix G discusses the limitations of our work and suggests potential avenues for future research.

## B VSD DATASETS

This section contains extended information of the VSD datasets used in our experiments.

### B.1 VSD FURNITURE DATASET

In this work, we employed the same EDS method from Barkan et al. (2023) for creating a novel VSD dataset in the Furniture domain. For a detailed description of the EDS method and protocol used to surface the query-candidate pairs for annotation, please refer to Sec. 3 and Barkan et al. (2023).

For the labeling task, four human annotators (two males and two females) were hired based on their expertise in furniture and home design domains. These experts are graduate students in architecture and design programs and have been working for several years as human annotators and data curators for machine learning systems in the fields of apparel, jewelry, furniture, home design, and artwork. The annotators were presented with a series of image pairs and tasked with determining whether each pair was similar or dissimilar based on their judgment. The primary guideline communicated to the annotators was that similar images (positive pairs) are those containing objects demonstrating visual similarity in terms of style, design, and relatedness, making them **suitable as alternative recommendations to each other from a visual perspective**.

Following the EDS protocol, image pairs were distributed among the experts with overlap, such that each image pair was labeled by at least two different experts to ensure consistency. Label disagreements were discussed to reach consensus, and unresolved pairs were excluded from the dataset (to avoid label collision during training).

Moreover, each annotator's self-consistency was measured to ensure integrity: For each annotator, a set of 200 pairs was randomly sampled, with each pair shown to the annotator at three different times. We then validated the annotator's consistency by comparing the annotation made at each time.

The resulting dataset, after removing duplicates and excluding controversial pairs, consists of 63,298 labeled query-retrieval pairs, with 39,194 labeled as positive and 24,104 as negative.

Examples from the VSD Furniture dataset are presented in Figures 2 and 3. These figures present examples of both positive and negative pairs. Positive pairs are identified as visually similar by our expert annotators, while negative pairs exhibit visual dissimilarity despite being ranked within the top-K candidates by one of the generator models. Therefore, these *hard* negatives are typically anticipated to pose a greater challenge for evaluation compared to randomly selected negatives.

The experts were compensated on an hourly basis, with annotation throughput ranging between 110 and 140 pairwise annotations per working hour, excluding controversial examples, which were addressed in designated discussion sessions. Each expert labeled approximately 35K image pairs due to overlapping sets. The annotation project required around 1,200 expert hours, with each expert contributing approximately 300 hours over four months, as they worked part-time.

The VSD Furniture dataset will be released under the permissive CC0 license as a self-contained package, including both the GUIE-Furniture images (which are also released under the CC0 license) and their pairwise labels. This ensures accessibility and long-term usability for the sake of reproducibility. Notably, the GUIE-Furniture dataset does not contain any confidential or sensitive data, and the same applies to the resulting VSD-Furniture dataset.

648
649
650
651
652
653
654
655
656
657
658
659
660
661
662
663
664
665
666
667
668
669
670
671
672
673
674
675
676
677
678
679
680
681
682
683
684
685
686
687
688
689
690
691
692
693
694
695

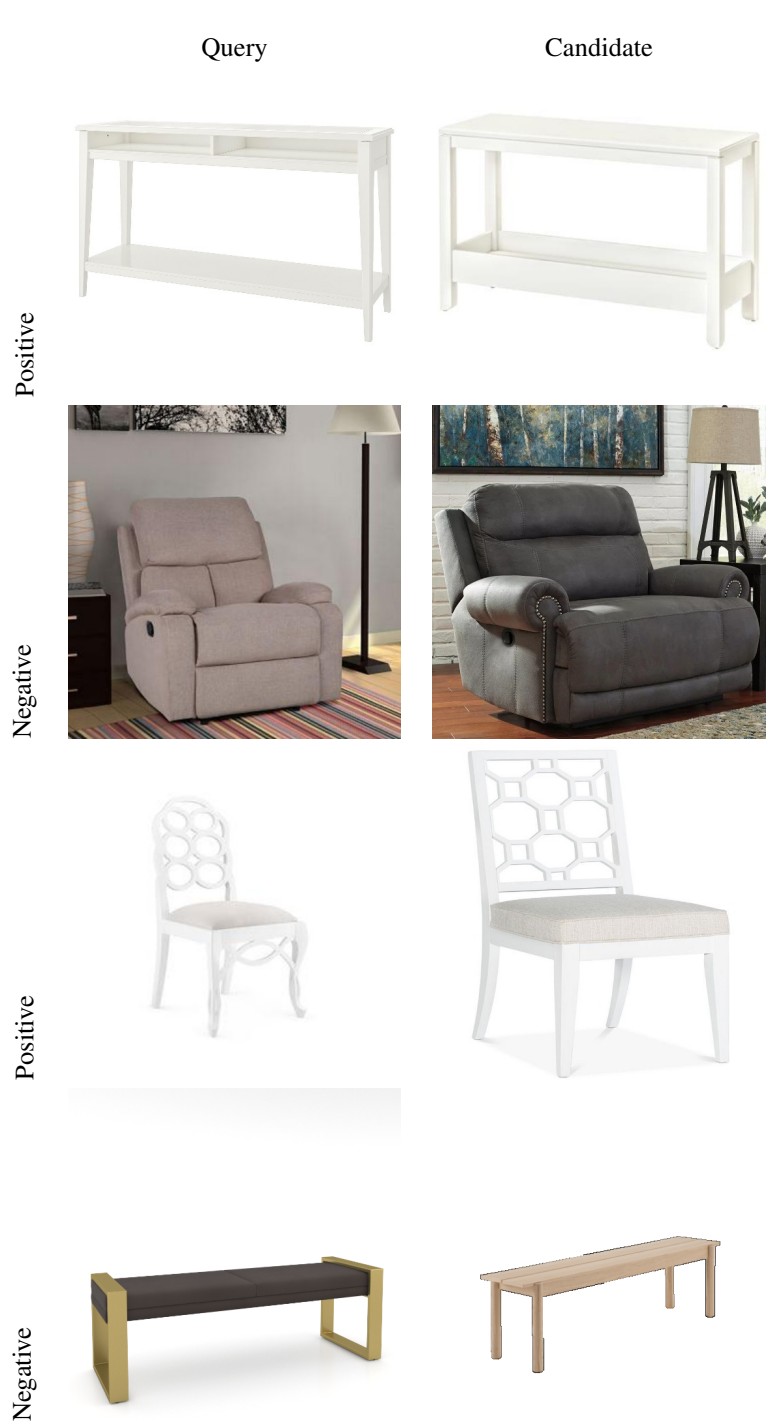

Figure 2: Labeled examples from the Furniture dataset. Each row displays the query image (left) and the candidate image retrieved from the top-K retrievals by one of the generator models (right). The label assigned by the experts for each pair is presented on the left side of the row.

696
697
698
699
700
701

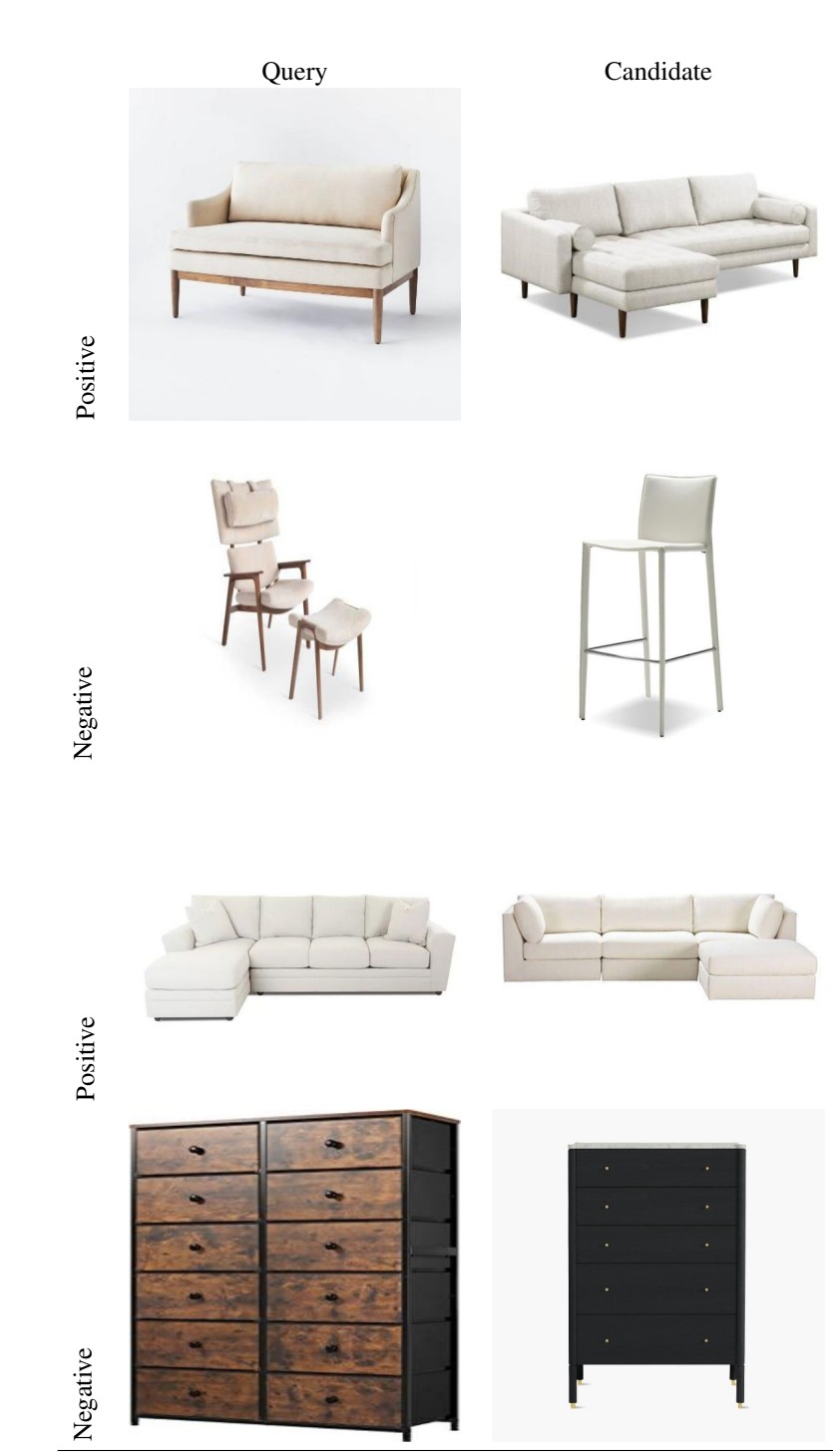

Figure 3: Labeled examples from the Furniture dataset. Each row displays the query image (left) and the candidate image retrieved from the top-K retrievals by one of the generator models (right). The label assigned by the experts for each pair is presented on the left side of the row.

## B.2 VSD FASHION DATASET

The VSD Fashion dataset was released by Barkan et al. (2023) for accelerating VSD research. In our experiments, we utilized the VSD Fashion dataset which is publicly available on HuggingFace[5]. This dataset was curated from a subset of the DeepFashion dataset Liu et al. (2016), comprising 52,712 images of clothing worn by models. According to Barkan et al. (2023), in order to construct this dataset, 2,000 query images were meticulously selected, and top-K candidate images were retrieved based on visual similarity according to the generator models. These candidates underwent expert annotation, resulting in a dataset featuring 54,170 positively or negatively labeled image pairs.

Compared to the Fashion dataset, our newly introduced Furniture dataset includes 75% more queries (3,494) than the Fashion dataset and a greater number of labeled image pairs. Moreover, our experimental results reveal that the Furniture dataset is significantly more challenging, as evidenced by the lower VSD metric scores achieved on this dataset relative to those on the Fashion dataset. This increased difficulty can be attributed to the presence of complex and varied distractors commonly found in furniture scenes, which make it more difficult for models to isolate and focus on the objects of interest. In contrast, the Fashion dataset primarily contains images of individuals modeling clothing in relatively clean and isolated environments, resulting in fewer distractors and an easier setting for visual similarity discovery.

## C SPA ABLATION STUDY

In this section, we present an ablation study supporting the specific SPA implementation used in this work.

### C.1 SOFT VS. NON-SOFT POSITIVE AUGMENTATION

To evaluate the necessity of label softness in SPA, we conducted an experiment by setting $\beta = 0$ in the positiveness function defined in Eq. 3. This adjustment effectively assigns a weight of 1 to all augmented positives, making them equivalent to the GT positives. We denote this non-soft variant of SPA as Non-Soft Positive Augmentation (NSPA). Table 3 presents a comparison of the performance of SupCon, SupCon$_{SPA}$, SupCon$_{NSPA}$, and the pretrained baseline (Pre).

As shown in Table 3, the use of NSPA significantly reduces the effectiveness of the augmented positives. In both the Furniture and Fashion domains, NSPA consistently underperforms relative to SPA, and in some cases, even falls behind SupCon and the pretrained baseline. These results indicate that relying solely on the GT labels often yields better performance than using SupCon$_{NSPA}$. This highlights the critical role of the positiveness function, which introduces softness to transitive augmented labels by assigning varying confidence levels during loss supervision.

### C.2 SPA PARAMETERS

The SPA method relies on two key parameters, $\beta$ and $L$, to control the positiveness scores of the augmented labels, as detailed in Section 6.1. We conducted an ablation study to determine the optimal values for these parameters, performing a grid search over $L$ (ranging from 2 to 7) and $\beta$ (ranging from 0.1 to 2.0), using DCS as the monitored evaluation metric. Based on validation set performance, the best results on average were achieved with $L = 7$ and $\beta = 0.7$. Results from a representative subset of experimental runs on the **test** set are presented in Table 4.

This study demonstrates that the test set performance closely aligns with that of the validation set, indicating that the selected parameter values ($L = 7$ and $\beta = 0.7$) generalizes well, yielding near-optimal results on the DCS metric for both datasets. While in this work we focused on monitoring the DCS metric, it is important to note that the choice of metric to monitor can vary depending on the user's preferences and specific requirements.

## D VSD METRIC CONSISTENCY TESTS ACROSS VARIOUS COVERAGE LEVELS

We define the coverage (**COV**) of a query for a model as the fraction of the query's top-5 retrieved results that are labeled, with COV ranging from 0 (zero coverage) to 1 (full coverage). For example, if three among the top-5 results retrieved by the model are labeled, the query has a COV of 0.6. A fully covered model has all its queries fully labeled.

---

[5]https://huggingface.co/datasets/vsd-benchmark/vsd-fashion

| Backbone | Method | Furniture | | | | | Fashion | | | | |
|---|---|---|---|---|---|---|---|---|---|---|---|
| | | EHR | | AUC | | DCS | EHR | | AUC | | DCS |
| | | @5 | @20 | mic | mac | | @5 | @20 | mic | mac | |
| SWAG | $SupCon_{SPA}$ | **88.19** | **83.67** | **77.54** | **74.8** | **69.78** | **99.37** | **98.8** | **86.63** | **92.79** | **87.89** |
| | $SupCon_{NSPA}$ | 80.9 | 79.1 | 69.67 | 69.9 | 66.71 | 99.19 | 98.64 | 85.37 | 91.76 | 84.29 |
| | SupCon | 83.92 | 80.79 | 76.27 | 74.4 | 68.48 | 99.19 | 98.52 | 83.65 | 92.22 | 85.19 |
| | Pre | 80.76 | 78.16 | 69.01 | 69.68 | 65.54 | 97.71 | 97.29 | 75.68 | 87.14 | 75.62 |
| OC | $SupCon_{SPA}$ | **88.23** | **83.76** | **78.27** | **75.35** | **69.97** | **99.41** | **98.86** | **86.48** | **92.6** | **87.81** |
| | $SupCon_{NSPA}$ | 84.29 | 81.6 | 71.35 | 71.53 | 67.57 | 99.3 | 98.68 | 85.23 | 91.69 | 83.86 |
| | SupCon | 84.84 | 81.38 | 77.03 | 74.74 | 68.57 | 98.93 | 98.48 | 83.53 | 92.06 | 87.06 |
| | Pre | 83.09 | 80.32 | 71.35 | 71.53 | 67.57 | 99.1 | 98.49 | 80.72 | 90.25 | 82.85 |
| DINO2 | $SupCon_{SPA}$ | **88.13** | **83.74** | **78.41** | **75.49** | **70.22** | **99.26** | **98.77** | **86.57** | **92.87** | **87.74** |
| | $SupCon_{NSPA}$ | 85.63 | 81.5 | 72.5 | 72.27 | 68.28 | 99.2 | 98.68 | 85.19 | 91.65 | 83.8 |
| | SupCon | 85.28 | 81.04 | 76.88 | 74.66 | 68.67 | 99.03 | 98.34 | 83.5 | 92.13 | 87.19 |
| | Pre | 85.14 | 80.38 | 72.38 | 72.27 | 67.65 | 97.3 | 96.87 | 70.1 | 86.14 | 69.42 |
| BEiT | $SupCon_{SPA}$ | **88.82** | **83.87** | **77.68** | **75.39** | **69.59** | **98.69** | **98.37** | **84.93** | **92.0** | **87.58** |
| | $SupCon_{NSPA}$ | 84.53 | 80.7 | 70.7 | 75.39 | 67.73 | 98.62 | 97.98 | 83.08 | 90.35 | 84.46 |
| | SupCon | 86.22 | 80.8 | 76.61 | 75.11 | 68.28 | 98.36 | 98.12 | 81.85 | 91.43 | 84.45 |
| | Pre | 84.53 | 80.18 | 70.7 | 72.15 | 67.73 | 96.03 | 95.43 | 75.67 | 87.85 | 83.85 |

Table 3: Soft vs. non-soft positive augmentation comparison. See section C.1 for more details.

In this experiment, we examine the correlation between VSD-metric evaluation results across various coverage levels and HR@5 results in the ideal full coverage scenario, which serves as a benchmark. This analysis assesses the reliability of each metric under different coverage conditions by comparing its results to HR@5 in the fully covered dataset. HR@5 is chosen as a benchmark because, in fully covered scenarios, it is one of the most widely used top-K metrics for evaluating IR systems in real-world applications. To explore this, we selected queries that were fully covered by each model and then artificially reduced their coverage by removing top-ranked annotations, simulating lower coverage scenarios, across various coverage levels.

The results for the Furniture and Fashion datasets are presented in Tables 5 and 6, respectively. We observe similar trends across both datasets: EHR exhibits the strongest correlation with HR@5 in the coverage range of 0.2 to 0.8, where DCS, AUC, and BPF are approximately on par. At full coverage (COV = 1), EHR naturally achieves perfect correlation with HR@5, as they coincide. However, EHR cannot be computed at 0% coverage.

Based on these findings, we conclude that in very low coverage scenarios, DCS and AUC are preferable due to their ability to maintain a positive correlation with HR@5 under full coverage conditions. While BPF also exhibits a positive correlation in low coverage settings, it underperforms compared to DCS and AUC.

# E IMPLEMENTATION DETAILS

For each backbone, the same architectural modification was uniformly applied across both self-supervised and supervised methods to ensure a consistent evaluation framework. Specifically, we modified each backbone by appending a **projection head** composed of the following layers:

1. A linear layer with input and output dimensionality equal to $feat_{dim}$, corresponding to the output dimensionality of the backbone's final layer.

2. A ReLU activation function.

3. A second linear layer with input and output dimensionality set to $proj_{dim}$.

We set $proj_{dim}$ to 2048 to standardize the projection dimensionality across all models. To compute the similarity between two images, we use the cosine similarity between the representations generated by the output of the projection head for supervised methods, and the representations produced by the finetuned backbone (i.e., the inputs to the projection head) for self-supervised methods[6].

---

[6]In SimCLR, these backbone representations are used for downstream tasks. Indeed, in our experiments, for self-supervised methods, these representations outperformed those produced by the output of the projection head.

| | | Fashion | | | | | Furniture | | | | |
|---|---|---|---|---|---|---|---|---|---|---|---|
| | | EHR | | AUC | | DCS | EHR | | AUC | | DCS |
| $L$ | $\beta$ | @5 | @20 | mac | mic | | @5 | @20 | mac | mic | |
| 2 | 0.10 | 84.86 | 81.32 | 74.9 | 77.17 | 68.75 | 99.13 | 98.64 | 92.23 | 83.95 | 87.35 |
| | 0.30 | 84.95 | 81.2 | 74.91 | 77.17 | 68.73 | 99.01 | 98.52 | 92.27 | 83.97 | 87.35 |
| | 0.50 | 84.7 | 81.39 | 74.85 | 77.14 | 68.69 | 98.96 | 98.66 | 92.23 | 83.88 | 87.35 |
| | 0.70 | 84.79 | 81.3 | 74.87 | 77.06 | 68.69 | 99.01 | 98.49 | 92.28 | 83.9 | 87.35 |
| | 0.90 | 84.38 | 81.15 | 74.83 | 77.04 | 68.69 | 99.07 | 98.66 | 92.33 | 83.89 | 87.35 |
| 3 | 0.10 | 84.96 | 81.79 | 74.59 | 77.39 | 69.39 | 99.26 | 98.86 | 92.61 | 84.78 | 87.39 |
| | 0.30 | 86.02 | 82.15 | 74.42 | 77.43 | 69.34 | 99.26 | 98.79 | 92.45 | 84.62 | 87.4 |
| | 0.50 | 86.98 | 82.44 | 74.24 | 77.25 | 69.27 | 99.2 | 98.7 | 92.33 | 84.54 | 87.42 |
| | 0.70 | 86.47 | 82.31 | 74.72 | 77.45 | 69.28 | 99.2 | 98.73 | 92.48 | 84.55 | 87.43 |
| | 0.90 | 86.78 | 82.31 | 74.9 | 77.53 | 69.28 | 99.19 | 98.68 | 92.33 | 84.4 | 87.43 |
| 4 | 0.10 | 84.97 | 81.63 | 73.37 | 76.17 | 69.26 | 99.06 | 98.84 | 92.49 | 85.76 | 87.31 |
| | 0.30 | 85.64 | 82.1 | 73.58 | 76.35 | 69.41 | 99.07 | 98.75 | 92.54 | 85.31 | 87.48 |
| | 0.50 | 86.23 | 82.36 | 73.93 | 77.19 | 69.6 | 99.05 | 98.78 | 92.64 | 85.41 | 87.58 |
| | 0.70 | 86.99 | 82.96 | 74.53 | 77.36 | 69.67 | **99.4** | 98.79 | 92.51 | 85.21 | 87.59 |
| | 0.90 | 87.61 | 83.16 | 74.61 | 77.39 | 69.55 | 99.15 | 98.76 | 92.71 | 85.18 | 87.54 |
| 5 | 0.10 | 83.52 | 80.7 | 72.85 | 74.9 | 68.95 | 98.82 | 98.74 | 92.38 | 86.41 | 86.8 |
| | 0.30 | 85.2 | 81.77 | 73.16 | 75.48 | 69.25 | 99.0 | **98.93** | 92.55 | 86.51 | 87.38 |
| | 0.50 | 86.55 | 82.01 | 74.22 | 77.13 | 69.68 | 99.12 | 98.79 | 92.59 | 86.36 | 87.61 |
| | 0.70 | 87.51 | 83.2 | 74.69 | 77.71 | 69.75 | 99.19 | 98.81 | 92.81 | 85.83 | 87.73 |
| | 0.90 | 87.6 | 83.18 | 74.79 | 77.72 | 69.67 | 99.28 | 98.77 | 92.69 | 85.51 | 87.7 |
| 6 | 0.10 | 83.39 | 80.43 | 71.81 | 72.91 | 68.18 | 99.17 | 98.92 | 92.2 | 86.49 | 86.12 |
| | 0.30 | 84.93 | 81.22 | 73.59 | 75.81 | 69.43 | 99.08 | 98.81 | 92.59 | 86.49 | 87.13 |
| | 0.50 | 87.33 | 82.73 | 74.91 | 77.29 | 69.84 | 99.18 | 98.76 | 92.6 | 86.57 | 87.68 |
| | 0.70 | 88.24 | 83.17 | **75.1** | **77.85** | 69.9 | 99.31 | 98.58 | **92.98** | 86.48 | 87.89 |
| | 0.90 | 87.89 | 83.47 | 74.86 | 77.8 | 69.74 | 99.34 | 98.89 | 92.81 | 85.81 | 87.8 |
| 7 | 0.10 | 82.32 | 79.84 | 71.65 | 72.44 | 68.31 | 99.3 | 98.88 | 91.88 | 86.06 | 85.41 |
| | 0.30 | 85.33 | 81.18 | 74.3 | 76.29 | 69.67 | 99.05 | 98.92 | 92.29 | 86.67 | 87.04 |
| | 0.50 | 87.27 | 82.87 | 74.96 | 77.53 | 69.79 | 99.12 | 98.91 | 92.47 | 86.5 | 87.71 |
| | 0.70 | **89.02** | **83.85** | 75.02 | 77.85 | **69.92** | 99.36 | 98.53 | 92.86 | **86.73** | **87.92** |
| | 0.90 | 88.32 | 83.43 | 74.51 | 77.64 | 69.76 | 99.11 | 98.86 | 92.78 | 86.13 | 87.9 |

Table 4: Ablation study on the positiveness function parameters $L$ and $\beta$ defined in section 6.1. See section C for details.

For optimization, we employed the AdamW optimizer with PyTorch's default settings, except for the learning rate. A cosine learning rate scheduler was used to decay the learning rate to a minimum of 10% of its initial value. Early stopping was applied based on performance on a VSD-relevant metric evaluated (DCS in this work) on the validation set, and the best-performing checkpoint was selected accordingly. Hyperparameters specific to each supervised and self-supervised method, as well as learning rate and batch size, were automatically tuned using the default TPE sampler in Optuna Akiba et al. (2019). Tuning was conducted independently for each combination of finetuning method, model architecture, and dataset, based on the validation split.

While our primary goal is to demonstrate the benefit of supervised finetuning on VSD labels, irrespective of the specific learning method, we followed best practices to ensure a fair comparison across the evaluated supervised methods (SupCon, TRPL, and Con). Specifically, we adopted the same preprocessing and data augmentation pipeline proposed in Khosla et al. (2021) for all supervised methods, and all methods utilized the same GT labels for supervision. In our experiments, we employed SupCon following Eq. 2 from Khosla et al. (2021), as this formulation has been shown to produce superior results. TRPL and Con losses were implemented as described in their original works, building upon the codebase released by Musgrave et al. (2020).

For self-supervised finetuning, we employed the **DINO** Caron et al. (2021) and **SimCLR** Chen et al. (2020) paradigms, using the recommended preprocessing and augmentation pipelines as proposed

---

This can be attributed to the fact that both SimCLR and DINO losses do not utilize VSD labels and therefore optimize for objectives that differ significantly from the VSD task.

| COV | Metric | r | KT | SC |
|-----|--------|------|------|------|
| 0.0 | BPF | 0.26 | 0.21 | 0.27 |
| 0.2 | BPF | 0.43 | 0.35 | 0.45 |
| 0.4 | BPF | 0.54 | 0.44 | 0.55 |
| 0.6 | BPF | 0.61 | 0.5 | 0.63 |
| 0.8 | BPF | 0.67 | 0.55 | 0.69 |
| 1.0 | BPF | 0.71 | 0.6 | 0.74 |
| 0.0 | DCS | 0.34 | 0.22 | 0.29 |
| 0.2 | DCS | 0.48 | 0.32 | 0.42 |
| 0.4 | DCS | 0.59 | 0.41 | 0.53 |
| 0.6 | DCS | 0.68 | 0.49 | 0.62 |
| 0.8 | DCS | 0.75 | 0.56 | 0.69 |
| 1.0 | DCS | 0.8 | 0.61 | 0.75 |
| 0.0 | EHR@5 | $nan$ | $nan$ | $nan$ |
| 0.2 | EHR@5 | 0.6 | 0.53 | 0.58 |
| 0.4 | EHR@5 | 0.77 | 0.7 | 0.75 |
| 0.6 | EHR@5 | 0.88 | 0.81 | 0.86 |
| 0.8 | EHR@5 | 0.95 | 0.91 | 0.94 |
| 1.0 | EHR@5 | 1.0 | 1.0 | 1.0 |
| 0.0 | $AUC_{mac}$ | 0.15 | 0.11 | 0.14 |
| 0.2 | $AUC_{mac}$ | 0.38 | 0.28 | 0.36 |
| 0.4 | $AUC_{mac}$ | 0.49 | 0.39 | 0.5 |
| 0.6 | $AUC_{mac}$ | 0.56 | 0.47 | 0.6 |
| 0.8 | $AUC_{mac}$ | 0.62 | 0.53 | 0.67 |
| 1.0 | $AUC_{mac}$ | 0.65 | 0.58 | 0.73 |

Table 5: The correlation between VSD-metric evaluation results computed under simulated low coverage conditions of the fully covered Furniture dataset and the benchmark HR@5 evaluation results obtained from the fully covered dataset. See Appendix D for details.

| COV | Metric | r | KT | SC |
|-----|--------|------|------|------|
| 0.0 | BPF | 0.08 | 0.1 | 0.11 |
| 0.2 | BPF | 0.2 | 0.15 | 0.17 |
| 0.4 | BPF | 0.32 | 0.2 | 0.23 |
| 0.6 | BPF | 0.36 | 0.23 | 0.27 |
| 0.8 | BPF | 0.43 | 0.29 | 0.35 |
| 1.0 | BPF | 0.51 | 0.37 | 0.44 |
| 0.0 | DCS | 0.28 | 0.16 | 0.2 |
| 0.2 | DCS | 0.47 | 0.14 | 0.18 |
| 0.4 | DCS | 0.65 | 0.28 | 0.34 |
| 0.6 | DCS | 0.72 | 0.34 | 0.41 |
| 0.8 | DCS | 0.79 | 0.4 | 0.49 |
| 1.0 | DCS | 0.91 | 0.52 | 0.62 |
| 0.0 | EHR@5 | $nan$ | $nan$ | $nan$ |
| 0.2 | EHR@5 | 0.35 | 0.46 | 0.47 |
| 0.4 | EHR@5 | 0.63 | 0.61 | 0.63 |
| 0.6 | EHR@5 | 0.79 | 0.73 | 0.74 |
| 0.8 | EHR@5 | 0.88 | 0.79 | 0.8 |
| 1.0 | EHR@5 | 1.0 | 1.0 | 1.0 |
| 0.0 | $AUC_{mac}$ | 0.2 | 0.11 | 0.12 |
| 0.2 | $AUC_{mac}$ | 0.45 | 0.32 | 0.34 |
| 0.4 | $AUC_{mac}$ | 0.55 | 0.4 | 0.43 |
| 0.6 | $AUC_{mac}$ | 0.59 | 0.49 | 0.52 |
| 0.8 | $AUC_{mac}$ | 0.62 | 0.56 | 0.61 |
| 1.0 | $AUC_{mac}$ | 0.69 | 0.74 | 0.79 |

Table 6: The correlation between VSD-metric evaluation results computed under simulated low coverage conditions of the fully covered Fashion dataset and the benchmark HR@5 evaluation results obtained from the fully covered dataset. See Appendix D for details.

in the original works. These configurations yielded better results than the configurations set for the supervised methods. Although these methods are unsupervised in nature and are typically used as pretraining steps prior to downstream finetuning, in this work, we utilize the VSD labels to guide the training process indirectly. To that end, we applied the same hyperparameter optimization and VSD metric monitoring procedures as used for the supervised learning methods, selecting the best-performing model checkpoint based on validation performance[7]. We found that applying DINO and SimCLR in a purely unsupervised setting, without hyperparameter tuning or metric-based monitoring, often results in performance inferior to that of the pretrained model.

It is important to emphasize that neither SimCLR nor DINO incorporates any VSD label information directly in their loss functions. Our goal in comparing with self-supervised finetuning is to investigate whether optimizing self-supervised objectives such as DINO and SimCLR can serve as stronger baselines than the pretrained models and yield representations that are effective for the VSD task, while leveraging VSD labels only indirectly through validation-based monitoring.

## F  METRICS

This section covers the different metrics we used in our experiments.

1. **Discounted Credit Score (DCS)** is explained in detail in Section 4, specifically we set $\alpha = 10$.

2. **Hit Ratio at K (HR@K)** (Krichene & Rendle, 2022) is explained in detail in Section 5.

3. **Estimated Hit Ratio at $K$ (EHR@K)** is explained in detail in Section 5.

4. **The Area Under the Receiver Operating Characteristic Curve - ROC-AUC (AUC)** (Fawcett, 2006) is introduced in Barkan et al. (2023) as a key performance metric for VSD tasks. ROC-AUC evaluates the model's capability to accurately distinguish between positive and negative observations. This is achieved by adjusting a threshold to plot true positive rates versus false positive rates on the ROC curve. Notably, ROC-AUC is computed in two distinct manners:

    (a) Micro-averaged AUC ($AUC_{mic}$), which aggregates the scores of all query-candidate pairs.
    (b) Macro-averaged AUC ($AUC_{mac}$), which calculates the AUC for each query individually followed by averaging these scores across all queries.

    This metric effectively measures the likelihood that a randomly selected positive pair is ranked higher by the model than a randomly selected negative pair, demonstrating the model's proficiency in prioritizing positive over negative data without considering their absolute positions in the ranking.

5. **BPREF (BPF)** (Buckley & Voorhees, 2004) is a metric used to evaluate the effectiveness of information retrieval systems when complete relevance judgments are unavailable, and as been shown to outperform mAP and Precision@K in such cases. It assesses how well a system ranks known relevant documents above known irrelevant ones. For each relevant document, BPF calculates the proportion of irrelevant documents ranked above it. The aim is for relevant documents to outrank all irrelevant ones, indicating superior retrieval performance. This metric offers a practical means to measure search result quality in large databases where full relevance assessments are not feasible. In our experiments we utilized a specific BPF implementation from the IR Measures package[8].

## G  LIMITATIONS AND FUTURE WORK

While our proposed methods and metrics demonstrate effectiveness in learning and evaluating VSD models, several limitations and avenues for future work warrant consideration. First, the applicability of DCS and EHR metrics may be influenced by specific dataset characteristics and task nature. Future work could investigate the adaptability of these metrics across diverse domains and benchmark

---

[7]The best checkpoint was selected from those obtained after the first training epoch onward (excluding the pretrained model itself), which means it is possible for the best checkpoint to perform worse than the original pretrained model.

[8]https://github.com/terrierteam/ir_measures

datasets to evaluate their generalizability to IR applications beyond VSD, including document retrieval and general search.

Second, although the SPA method improved VSD performance on the datasets used in this work, its effectiveness may depend on the quality of the GT labels. Investigating SPA's robustness under noisy labeling conditions, such as datasets with a high proportion of incorrect or ambiguous annotations, remains an open question for future research. Moreover, since SPA is heuristic in nature and validated empirically, future work could seek to establish a theoretical grounding for the method, explore its properties when integrated with different loss functions, and examine its implications for positive augmentation across various scenarios, including those involving noisy GT labels.

Lastly, while our proposed methods have been evaluated on datasets of considerable size, their scalability to even larger datasets and real-world applications (e.g., e-commerce platforms with millions of items) remains an open avenue for future research.

We believe that addressing the aforementioned limitations and exploring these avenues for future work will contribute to the ongoing advancement and applicability of our proposed metrics and methods in the evolving landscape of VSD research and beyond.