# OpenReview forum: "Learning and Evaluating Visual Similarity Discovery under Incomplete Labeling"
_ICLR.cc/2026/Conference — ICLR 2026 Conference Withdrawn Submission_

### Official Review · Reviewer_fBez · 2025-10-31

**Soundness:** 2
**Presentation:** 2
**Contribution:** 2
**Rating:** 2
**Confidence:** 4

**Summary:**

This paper addresses the challenge of Visual Similarity Discovery (VSD) under incomplete labeling, where many visually similar image pairs remain unlabeled due to generator-based retrieval bias. This limitation affects both evaluation reliability and model training.

The paper makes three main contributions:

- VSD-Furniture Dataset:
A new human-annotated dataset of over 63K image pairs in the furniture domain, extending prior VSD benchmarks beyond fashion using the Efficient Discovery of Similarities (EDS) paradigm.

- Evaluation Metrics for Incomplete Labeling:
The authors propose two new metrics to improve robustness under sparse annotations:
Discounted Credit Score (DCS) emphasizes top-ranked retrievals and mitigates AUC’s limitations, while Estimated Hit-Ratio at K (EHR@K) normalizes for unlabeled results to ensure consistent evaluation.

- Supervised Fine-tuning with Soft Positive Augmentation (SPA):
Fine-tuning pretrained models such as CLIP, DINOv2, and BEiT on VSD labels significantly improves performance. The proposed SPA method further enhances results by inferring soft positive relationships among unlabeled pairs via weighted graph transitivity.

Overall, the paper presents a unified and practical framework for learning and evaluating VSD models in realistic, partially labeled settings.

**Strengths:**

- Clearly explains the necessity and significance of the Visual Similarity Discovery (VSD) task and its differences from traditional retrieval or recognition settings

- Provides comprehensive experiments with multiple pretrained models (CLIP, DINO, BEiT, etc.) and proposes quantitative evaluation using diverse metrics (AUC, BPREF, DCS, EHR@K).

- Introduces practical methods (DCS, EHR@K, SPA) that effectively address the issue of incomplete labeling and improve fine-tuning results.

**Weaknesses:**

- The main contribution is unclear. It combines dataset extension, metric design, and fine-tuning without a single coherent focus. The motivation for creating a new furniture-domain dataset using the same EDS pipeline is not fully convincing, as it seems an incremental domain extension.

- The logical flow around DCS is confusing. The paper links generator bias (EDS limitation) with AUC’s triplet limitation but does not clearly explain how DCS specifically resolves these issues.

- Presentation clarity is limited. The paper lacks a figure illustrating the full motivation and pipeline, and supervised fine-tuning is not novel—SPA should be emphasized more as the true methodological contribution.

**Questions:**

- Could the authors include a figure or diagram illustrating the overall motivation and pipeline, showing how dataset construction, metric design, and SPA are connected?

- Beyond supervised fine-tuning, have the authors considered alternative learning approaches more suited for VSD, such as semi-supervised, contrastive, or retrieval-specific adaptation methods?

- The paper mentions using multiple generator models to aggregate top-K retrievals—could the authors clarify how this aggregation is performed (e.g., union, weighted ranking, or score fusion)?

---

> ### Author Response · Authors · 2025-11-14
> **Authors response**
>
> 1. We thank the reviewer for the comment. However, we are unsure why the main contribution is considered unclear, especially since the reviewer’s own summary aligns well with what we explicitly and precisely state in Lines 85–89:
> (1) the introduction of a new dataset,
> (2) the proposal of two evaluation metrics for VSD, and
> (3) empirical evidence showing that supervised fine-tuning on VSD labels improves performance, with an additional boost from incorporating SPA.
>
> Regrading question 1: These three are interconnected components that together form a coherent and holistic contribution. The work advances VSD research on two complementary fronts:
>
> Algorithmic: through the introduction of SPA, which improves supervised fine-tuning.
>
> Evaluation: through the new VSD-Furniture dataset and improved VSD metrics.
>
> With regard to the reviewer’s comment that the furniture-domain dataset constitutes only an incremental extension, we respectfully disagree. The furniture domain represents a completely different visual domain from clothing, with distinct stylistic, geometric, and structural properties. Moreover, using the EDS pipeline is the only practical and cost-effective method currently known in the literature for constructing VSD datasets with reliable perceptual similarity labels, and this methodology has already been acknowledged by the vision community.
>
> For these reasons, we are uncertain how to interpret the statement that the motivation is “not fully convincing.”
>
>
>
> 2. We thank the reviewer for pointing out this concern in weakness 2. First, we would like to emphasize that DCS consistently outperforms AUC across a wide range of standard consistency tests, as demonstrated in Table 1 (see Sections 7.2.1 and 7.2.2). These results provide strong empirical evidence of DCS’s effectiveness relative to AUC.
> Second, we kindly refer the reviewer to the discussion in Lines 169–174, where we explain the limitation of AUC—specifically, its reliance on relative ranking within triplets while ignoring the absolute retrieval rank, a critical factor when evaluating perceptual similarity. In contrast, as detailed in Lines 175–207, DCS directly addresses this weakness by explicitly incorporating the absolute rank of retrieved items. This allows DCS to better capture meaningful differences in retrieval quality, especially in scenarios where the absolute ordering is essential.
> Together, these theoretical motivations and empirical results clarify how DCS resolves the issues identified with AUC and why it serves as a more robust and informative metric in the context of VSD.
>
>
> 3. We thank the reviewer for these comments in weakness 3 and questions 2 and 3. Regarding the clarity of the presentation, we kindly refer the reviewer to the Introduction, which provides the full motivation behind all contributions in the paper. The overall pipeline is described explicitly in Lines 34–40. Due to space limitations, we could not include an additional figure, yet, the process is explained in a clear and straightforward manner in just a few sentences.
>
> While supervised fine-tuning itself is not novel, we never claim it as a contribution. Our contribution lies in demonstrating empirically that applying existing supervised fine-tuning methods to VSD-specific labels significantly improves performance (under-explored in previous VSD works). This provides evidence that deep models are capable of learning nuanced perceptual visual similarity that aligns with expert annotations. The novel methodological contribution is indeed SPA, as clearly stated in Lines 76–89, and we agree that SPA is the algorithmic innovation of the paper.
>
> Regarding the reviewer’s question about alternative learning approaches: The methods we employ - SupCon, Triplet, and Contrastive Loss - are all inherently contrastive in nature and well-suited for similarity learning. Exploration of semi-supervised or other retrieval-specific adaptation approaches is indeed interesting, but it falls outside the scope of the current work, which explicitly focuses on supervised learning using high-quality perceptual similarity labels.
>
> Regarding the aggregation method: We used the union of the top-K retrievals from all generator models.
>
> We appreciate the reviewer’s thoughtful suggestions and feedback. Thank you.

---

### Official Review · Reviewer_DEhh · 2025-10-31

**Soundness:** 3
**Presentation:** 3
**Contribution:** 3
**Rating:** 6
**Confidence:** 3

**Summary:**

The paper introduces a dataset created by experts on the topic of VSD. VSD (Visual Similarity Discovery) is the task of finding objects that are similar to the query object, but preferably not the exact same object. This is very relevant for the task of product suggestions, which we find in all commerce sites. They highlight the problem of incomplete labeling when training models for these tasks. In addition to the dataset, propose two evaluation metrics designed for VSD in the case of missing labels, and lastly empirical evidence showing that fine-tuning on VSD datasets significantly improves performance.

**Strengths:**

The authors address a very real and important problem in the VSD literature, namely incomplete labels in a VSD dataset. Furthermore, the authors have performed rigorous testing of the proposed metrics and show that they are "better" when using them on VSD with incomplete labeling.

**Weaknesses:**

"Using cosine similarity alone to generate candidate pairs systematically biases the dataset toward the geometry of the pretrained embedding space, which may not reflect perceptual or semantic similarity. The Circle-loss paper explicitly shows why cosine similarity is an incomplete measure of true pair similarity. Suggestions on what else to do could be to fine-tune the embedding models using the circle-loss, or similar. Other approaches might be to combine or use an ensemble of similarity metrics. Also, a minor formatting error: overlapping text in figure and paragraph on line 206-207.

\cite{https://arxiv.org/abs/2002.10857}

**Questions:**

I would like to see a test of the use of the similarity metrics, or at least a discussion as to why you think solely using cosine-similarity is enough.

---

> ### Author Response · Authors · 2025-11-14
> **Authors response**
>
> We thank the reviewer for the insightful comment and constructive suggestions. We fully agree that approaches such as Circle Loss may offer valuable future directions for VSD research. However, incorporating such experiments is beyond the scope of the current work, and due to space limitations we cannot include additional training methodologies at this stage. We will explicitly mention Circle Loss and related alternatives as future research avenues, particularly in the context of the limitations of using cosine similarity.
>
> Regarding the request for a discussion of why we believe using cosine similarity alone is sufficient for constructing the dataset, our reasoning is straightforward. First, cosine similarity is widely used in the literature for retrieval and similarity tasks. More importantly, when cosine similarity was used within the EDS method, it consistently surfaced true positive perceptual similarities, which were subsequently verified by expert annotators. The dataset was ultimately constructed based on these expert validations, ensuring that retrieved image pairs indeed reflect genuine perceptual similarity, independent of any specific embedding geometry.
>
> It is also important to clarify that the losses used in our fine-tuning experiments are not limited to cosine-based formulations. They rely on the dot product without normalization, as in SupCon loss, and the same applies to the triplet and contrastive losses. Our goal is not to introduce a new loss function but to demonstrate that supervised fine-tuning, with or without SPA, consistently improves VSD performance across metrics and datasets, and to highlight the effectiveness of SPA across different loss functions.
>
> We appreciate the reviewer’s suggestion and believe it provides valuable perspective for future extensions of this work. Thank you!

---

### Official Review · Reviewer_XbZo · 2025-11-01

**Soundness:** 2
**Presentation:** 3
**Contribution:** 2
**Rating:** 4
**Confidence:** 4

**Summary:**

This paper presents a study on Visual Similarity Discovery (VSD)—retrieving visually similar but non-identical items—through a new dataset (VSD-Furniture), two evaluation metrics for handling incomplete labeling (Discounted Credit Score and Estimated Hit-Ratio@K), and a fine-tuning strategy called Soft Positive Augmentation (SPA). The authors position VSD as distinct from visual search or duplicate detection, focusing on perceptual similarity beyond exact matches. Experiments on the proposed dataset and fashion benchmarks show consistent gains from supervised fine-tuning and SPA.

**Strengths:**

The paper is well-written. It identifies a gap between standard visual search and perceptual similarity discovery. The newly proposed metrics try to mitigate some limitations of existing metrics—how incomplete labeling can bias retrieval evaluation—and the empirical validation may be technically right.

**Weaknesses:**

The main conceptual concern lies in the unclear distinction between visual similarity discovery and conventional visual search. From a technical standpoint, the two tasks share almost identical pipelines, and a standard visual search system can naturally serve as a discovery engine by filtering out identical products. They share a similar goal, i.e., to rank the most visually similar objects higher based on retrievals.

This weakens the motivation for defining VSD as a separate problem space. The necessity of introducing a new dataset becomes questionable, considering strong benchmarks such as Stanford Online Products and In-Shop Clothes Retrieval–among others–already exist for evaluating visual similarity and retrieval models. The proposed VSD-Furniture dataset is limited to a single product category and modest in scale, which restricts its generalizability and practical impact.

Furthermore, while the proposed metrics and fine-tuning strategies yield some improvements, the gains appear limited. Standard metrics such as Recall@K can already provide reasonable ranking–use different K values–estimates even under partial labeling, reducing the necessity of introducing new ones. Moreover, the reported benefits from fine-tuning are modest and do not seem to be significant observations.

**Questions:**

Claiming MAP is “less effective” or should be excluded entirely seems to be a stretch. It could be better if the authors provided stronger evidence, rather than relying on a single reference to justify this decision.

---

> ### Author Response · Authors · 2025-11-14
> **Authors response**
>
> 1. We thank the reviewer for raising this conceptual point. While we understand that visual search and visual similarity discovery (VSD) may appear similar at first glance, there is a fundamental distinction. Visual search is not restricted to suggesting different items; it often retrieves identical items from different viewpoints, as this is central to many retrieval applications. In contrast, VSD—by definition—focuses specifically on retrieving positives that represent different and distinct objects exhibiting high perceptual similarity, as we explicitly state in the abstract (“retrieving positives: images of distinct objects that exhibit perceptual similarity to a given query”) and reaffirm in the introduction (“the interest is on retrieving images of different and distinct items … that still share a high degree of perceptual similarity as judged by humans”).
>
> A key technical limitation of existing datasets such as Stanford Online Products and In-Shop Clothes Retrieval is that they do not provide similarity labels, which is at the heart of our study. These datasets only provide instance-level labels for identifying identical products, not human-assessed perceptual similarity between different items. Therefore, they cannot be used as benchmarks for the specific VSD task we address, which fundamentally requires perceptual similarity annotations.
>
> Both the Fashion dataset (introduced previously for the VSD task and acknowledged by the vision community) and the newly proposed VSD-Furniture dataset address precisely this gap: they provide pairwise similarity labels for images that do not depict the same object, and they enable the evaluation of models on true perceptual similarity, rather than product matching.
>
> Regarding the reviewer’s concern about the scale and category focus of the VSD-Furniture dataset, we note that the primary contribution is the availability of human perceptual similarity labels, which were previously missing for this task. While the dataset focuses on a single category, it complements the Fashion dataset and extends perceptual-similarity evaluation into a new domain.
>
> In summary, although visual search and VSD share similarities in pipeline structure, their objectives and required supervisory signals are fundamentally different. Because existing retrieval datasets lack perceptual similarity labels, they cannot serve as benchmarks for the VSD task. This is the core motivation for defining VSD as a distinct problem space and for introducing the new dataset. We appreciate the reviewer’s comment and hope this clarification helps highlight the necessity and relevance of our contributions.
>
>
> 2. We respectfully disagree with the reviewer’s statement that Recall@K provides reasonable ranking estimates under incomplete or partially labeled datasets. As noted in prior studies, Recall@K is well-known to be limited in such scenarios, and Precision@K (equivalent to HR@K in our work) suffers from the same issue due to missing labels. This limitation is precisely why earlier metrics such as BPREF and AUC were proposed as more suitable tools for evaluating information retrieval systems in the presence of incomplete annotations. Our proposed metrics, EHR and DCS, were designed to address this same challenge and, as shown in the experiments, demonstrate better consistency and robustness across a variety of incomplete-label conditions.
>
> Regarding the reviewer’s comment on the benefits of fine-tuning, we again respectfully disagree. The improvements obtained from fine-tuning without SPA are significant in almost all cases compared to the pretrained model. Moreover, once SPA is applied, the gains become substantially larger. As we explicitly explain in Lines 470–473, this provides empirical evidence that the annotated similarity labels are of high quality and that vision models are indeed capable of learning and generalizing human-perceived visual similarity.
>
> Regarding MAP, multiple previous studies, including the one presented BPREF have shown that it is less suitable for evaluating scenarios with incomplete labels. For this reason, we did not include MAP in our evaluation.
>
> Thank you for your feedback.

---

### Official Review · Reviewer_S4Wq · 2025-11-01

**Soundness:** 2
**Presentation:** 1
**Contribution:** 2
**Rating:** 2
**Confidence:** 4

**Summary:**

This paper focuses on Visual Similarity Discovery under scenarios with incomplete labeling and presents four core contributions: (1) constructing the VSD-Furniture dataset, a VSD dataset in the furniture domain; (2) designing two evaluation metrics—Discounted Credit Score (DCS) and Estimated Hit-Ratio at K to adapt to incomplete labeling; (3) verifying that supervised finetuning using VSD labels can improve model performance; and (4) proposing the Soft Positive Augmentation (SPA) method, which mines potential similarity relationships in unlabeled samples via weighted graph transitivity to further enhance finetuning effects. Experiments on the VSD-Furniture and Fashion datasets validated the consistency of the proposed metrics and the effectiveness of the finetuning methods.

**Strengths:**

1. Visual Similarity Discovery is a scientifically significant problem. It addresses a core demand in practical applications such as e-commerce and visual search.

**Weaknesses:**

1. **Notation and Formatting Issues:** The manuscript appears to be very hastily written, with numerous formatting and notation flaws. For instance, in the Abstract, there is ambiguity about an extra quotation mark after the phrase "retrieving positives". Additionally, the title of Figure 1 overlaps with the body text, making it difficult to read. These issues hinder the clarity of the manuscript and suggest a lack of careful proofreading. The authors should carefully polish and revise the manuscript to fix these formatting and notation problems.
2. **Outdated Experimental Comparisons:** The methods used in the experiments are relatively outdated, and the paper lacks comparisons with more recent VSD-related methods developed in recent years.
3. **Weak Theoretical Basis for Metrics:** The proposed metrics are only validated experimentally, with no proof of key mathematical properties. This lack of theoretical guarantees casts doubt on the metrics’ generalizability to other domains.

**Questions:**

Please refer to the "Weaknesses" section for relevant questions and suggestions.

---

> ### Author Response · Authors · 2025-11-14
> **Authors response**
>
> 1. We thank the reviewer for pointing out the formatting and notation issues. Regarding the “ambiguity” mentioned in the Abstract, we carefully re-examined the text but could not identify the “extra” quotation mark the reviewer refers to. We also appreciate the comment about the slight title overlap in Figure 1 and have corrected this issue. While we fully acknowledge the importance of clear formatting, we respectfully note that characterizing the manuscript as containing “numerous” flaws that hinder its clarity, based on what appears to be a single minor formatting issue (as also acknowledged by Reviewer DEhh) may overstate the extent of the problem. Nevertheless, we took this comment seriously and have thoroughly proofread the manuscript again to ensure that any remaining formatting or notation inconsistencies are addressed. Thank you.
>
>
> 2. We appreciate the reviewer’s concern regarding the experimental comparisons. Our experiments, however, are already extensive and include eight different backbones from recent years. Both the proposed metrics and the SPA method are validated across all of these backbones as well as on two recent and novel datasets specifically designed for the VSD task.
>
> The reviewer states that the paper lacks comparisons with “VSD-related methods developed in recent papers,” yet no specific methods or citations are provided. Without concrete references, it is difficult for us to address this concern in a targeted and meaningful way.
>
> Furthermore, given the breadth of our evaluation, we believe that its value does not come from continually “chasing” every new backbone/method that is released. Instead, the strength of our study lies in demonstrating the effectiveness of SPA across a diverse range of backbones and loss functions, and in showing that the proposed metrics are robust, consistent, and outperform existing alternatives on the VSD task through a series of experiments examining multiple angles and settings.
>
>
> 3. We thank the reviewer for raising this point. First, EHR@K is an unbiased metric with respect to HR@K. Second, DCS is thoroughly motivated and contrasted with AUC, with its behavior clearly illustrated in Figure 1. It is important to emphasize that this work is primarily empirical rather than theoretical, and therefore we focus on validating the proposed metrics through an extensive set of experiments. These evaluations include multiple robustness tests (see Table 1 and Tables 5 and 6 in the appendix), demonstrating the metrics’ effectiveness across varied conditions. While further investigation of theoretical properties would indeed be valuable, we consider this an interesting direction for future work.
>
> The reviewer also “casts doubt on the metrics’ generalizability to other domains.” Our work specifically targets visual similarity discovery, where the goal is to assess perceptual visual similarity. Although we believe the proposed metrics may potentially be applicable to additional modalities such as text or audio, exploring such extensions is outside the scope of this vision-focused study. We appreciate the reviewer’s suggestion and plan to investigate these broader theoretical and cross-domain aspects in future research. Thank you.

---

### Note · Authors · 2025-11-14

**Comment:**

We thank all reviewers for the time and effort invested in evaluating our paper, as well as for their comments and suggestions. While we addressed all concerns raised in our detailed responses, we nonetheless feel that the contribution and overall value of our work have been significantly underrated, particularly given its multiple substantive contributions to VSD research and its substantially more comprehensive, rich, and advanced treatment of VSD (on multiple fronts) compared to prior studies.

For these reasons, we have decided to withdraw the manuscript and submit it to another venue. We sincerely thank the reviewers again for their feedback.

**Withdrawal Confirmation:**

I have read and agree with the venue's withdrawal policy on behalf of myself and my co-authors.